# Isolation and Identification of Salinity-Tolerant Rhizobia and Nodulation Phenotype Analysis in Different Soybean Germplasms

Tong Yu [1,†], Xiaodong Wu [2,†], Yunshan Song [1], Hao Lv [1], Guoqing Zhang [1], Weinan Tang [1], Zefeng Zheng [1], Xiaohan Wang [1], Yumeng Gu [1], Xin Zhou [1], Jianlin Li [1], Siyi Tian [1], Xiuming Hou [1], Qingshan Chen [1], Dawei Xin [1,*] and Hejia Ni [1,*]

1   Key Laboratory of Soybean Biology of the Chinese Ministry of Education, Key Laboratory of Soybean Biology and Breeding, Genetics of Chinese Agriculture Ministry, College of Agriculture, Northeast Agricultural University, Harbin 150036, China; yutong6488@163.com (T.Y.); 18800426482@163.com (Y.S.); lv1055496656@163.com (H.L.); 17745156274@163.com (G.Z.); twn1229284249@163.com (W.T.); a01190455@neau.edu.cn (Z.Z.); 18246443958@163.com (X.W.); guyumeng7381@163.com (Y.G.); 13292395386@163.com (X.Z.); 15247991509@163.com (J.L.); tsynly@163.com (S.T.); 17835062485@163.com (X.H.); qshchen@126.com (Q.C.)
2   Heilongjiang Green Food Science Research Institute, Harbin 150000, China; 13081767953@163.com
*   Correspondence: dwxin@neau.edu.cn (D.X.); h10081@neau.edu.cn (H.N.)
†   These authors contributed equally to this work.

**Abstract:** Increasing the soybean-planting area and increasing the soybean yield per unit area are two effective solutions to improve the overall soybean yield. Northeast China has a large saline soil area, and if soybeans could be grown there with the help of isolated saline-tolerant rhizobia, the soybean cultivation area in China could be effectively expanded. In this study, soybeans were planted in soils at different latitudes in China, and four strains of rhizobia were isolated and identified from the soybean nodules. According to the latitudes of the soil-sampling sites from high to low, the four isolated strains were identified as HLNEAU1, HLNEAU2, HLNEAU3, and HLNEAU4. In this study, the isolated strains were identified for their resistances, and their acid and saline tolerances and nitrogen fixation capacities were preliminarily identified. Ten representative soybean germplasm resources in Northeast China were inoculated with these four strains, and the compatibilities of these four rhizobium strains with the soybean germplasm resources were analyzed. All four isolates were able to establish different extents of compatibility with 10 soybean resources. Hefeng 50 had good compatibility with the four isolated strains, while Suinong 14 showed the best compatibility with HLNEAU2. The isolated rhizobacteria could successfully establish symbiosis with the soybeans, but host specificity was also present. This study was a preliminary exploration of the use of salinity-tolerant rhizobacteria to help the soybean nitrogen fixation in saline soils in order to increase the soybean acreage, and it provides a valuable theoretical basis for the application of saline-tolerant rhizobia.

**Keywords:** isolation of rhizobia; resistance identification; nitrogenase activity; host compliance; nitrogen-fixing bacteria; soybean

## 1. Introduction

Soybean, as a kind of grain and oil, feeds diversified economic crops [1] and it is an important import and export material. With the advancement of the consumption structure of the population, the demand for soybeans in China is increasing year by year [2]. Nitrogen, which cannot be absorbed directly by plants [3], is one of the important factors that affects soybean yields [4]. Nitrogen fertilizers are mainly applied to increase crop yields in agricultural production. The excessive application of nitrogen fertilizer can cause soil nutrient imbalances, leading to the depletion of other important minerals

in the soil, such as calcium, phosphorus, and magnesium, and can even lead to soil acidification [5]. The symbiosis with rhizobia can provide a stable nitrogen source [6,7], improving yields while reducing pollution and costs [8]. Rhizobia transform nitrogen in the atmosphere into ammonia, which can be absorbed by plants, and this process takes place in the nodule organs. Auxin has three main functions during nodule development: cell cycle control, vascular tissue differentiation, and rhizobial infection [9]. Studies have reported the mechanisms of legume-nodulating bacteria (LNB) to improve soil health and crop yields. LNB can improve the utilization rate of nitrogen and other plant nutrients in soil by simulating the synthesis of plant hormones, thereby accelerating crop growth and development [10,11].

In recent years, rhizobium inoculation has become a hit in the process of growing soybeans [12], and it has been promoted in countries such as Brazil, the United States, and Argentina to improve soybean crop yields [13,14] and reduce the chemical fertilizer inputs and cost of soybean production with microbial nitrogen fixation [15], making them the main soybean exporters. In the 100-year history since the discovery of rhizobia, countries around the world have promoted their isolation and put studies on host specificity into practice [16,17]. For example, the encapsulation method has been used to improve the inoculation efficiency [18]. Related studies have played a role in advancing the development, promotion, and utilization of rhizobium agents [19]. Instead of the chemical fertilizer Phonska, 400 cc/kg granule and 600 cc/kg powder rhizobial isolates can be used in soybean planting [20]. $Fe_3O_4$ nanoparticles (NPs) and Rhizobium inoculation in common beans can improve the nodulation and nitrogen fixation during vegetative growth [21]. However, there is currently a large research gap in the application of salinity-tolerant rhizobacteria. In this study, soybeans were inoculated with acid-tolerant, salt-tolerant, and alkali-tolerant rhizobacteria to characterize their compatibilities and nitrogen fixation abilities, and to explore new application directions for salt-tolerant rhizobacteria.

Although the use of rhizosphere bacteria as a substitute for chemical nitrogen fertilizer to increase crop yields has significant application potential, there are substantial differences in the cultivated varieties and host specificities of rhizosphere bacteria in different regions, which limit their wide application [22]. The compatibility difference between leguminous plants and rhizosphere bacteria leads to an unstable nitrogen fixation efficiency and a poor rhizosphere bacteria inoculation effect, which limit the large-scale spread of rhizobia microflora. To improve the utilization of rhizobia in the field, it is necessary to obtain strains with lower host specificities, wide adaptations, and high nitrogen fixation efficiencies according to the local conditions. Screening rhizobia and soybean cultivars with high compatibilities can effectively improve the nitrogen fixation efficiencies of rhizobia, thereby increasing soybean yields.

As the largest soybean-growing area in China, Heilongjiang Province has the largest grain production capacity and six cumulative temperature zones [23], which make it suitable for the growth of various soybean cultivars [24,25]. However, a large part of the land in Heilongjiang Province is saline–alkali soil. In order to improve soybean yields by using saline–alkali-tolerant rhizobia in saline–alkali soil, rhizobia need to have various characteristics and be able adapt to harsh farmland environments. This study provides the multiple characteristics of the isolated and identified rhizobia, as well as their inoculation efficiencies on different soybean germplasm resources. This study provides an important theoretical basis for the application of rhizobia to improve soybean yields on saline–alkali land in Heilongjiang Province.

## 2. Materials and Methods

### 2.1. Isolation and Characterization of Rhizobia from Different Regions

The soybean cultivar Suinong 14 was used as the arrest host, sterilized via chlorine fumigation, and sown into soils isolated from Heilongjiang Province (43.26°~53.33° N, 121.11°~135.05° E), Shaanxi Province (31.95°~39.01° N, 105.29°~111.15° E), Yunnan Province (21.8°~29.15° N, 97.31°~106.11° E), and Guangdong Province (20.13°–25.31° N, 97.31°–106.11° E,

20.13°–25.31° N, and 109.39°–117.19° E). The soybean cultivation conditions were 25 °C and 16/8 h for the 25-day incubation. The roots were rinsed with water, and then full and bulky nodules were picked with forceps and placed in Petri dishes. The fresh rhizobia were surface-sterilized with sterile water and anhydrous ethanol under aseptic conditions and were then cut using a sterilized scalpel. The inner cut surface was affixed to the yeast juice agar (YMA) Congo-red solid medium for delineation and then incubated in a constant-temperature incubator at 28 °C. According to the surface polysaccharide morphologies of the rhizobia, rhizobia with milky-white or white-jam morphologies were picked and continuously streaked and purified until monoclones appeared. The purified monoclonal colonies obtained were streaked onto the YMA Congo-red solid medium and incubated at 28 °C for two days. Then, the sizes, shapes, edges, hardnesses, transparencies, and other apparent properties of the single colonies were observed [26].

The isolated monoclonal strains were cultured in a tryptone-yeast-extract liquid medium (TY liquid medium) on a shaker (28 °C, 2000 r·min$^{-1}$) until the OD$_{600}$ was 0.6. Bacterial genome DNA extraction was performed with a bacterial genome DNA extraction kit (Tiangen Biochemical Technology Beijing Co., Ltd., Beijing, China) according to the manufacturer's instructions. The conserved regions of the 16S rRNA genes of several bacterial fluids were amplified (1500 bp) with bacterial universal primers (F: AGAGTTTGATCCTG-GCTCAG; R: AAGGAGGTGATCCAGCC), and the amplified fragments were isolated for sequencing. The identification results were compared with the reported gene sequences in the NCBI by BLAST, and then MAGE was applied to construct a phylogenetic tree to determine their relative positions in the rhizobium system.

## 2.2. Identification of Rhizobium Resistance

The stress conditions of antibiotic, salinity, acid, and alkali stresses were selected to identify the rhizobium resistance. The rhizobia were cultured with a solid medium containing three different concentrations of antibiotics: 20 μg/L, 50 μg/L, and 100 μg/L. Rhizobia were cultured with the TY solid medium with NaCl concentrations of 0 mmol/L, 50 mmol/L, 100 mmol/L, and 150 mmol/L [27]. The pH of the medium was adjusted with HCl and NaOH to configure the TY solid medium with pH values from 5 to 10 to culture the rhizobia [28]. The growth statuses of the strains were observed after 5 days of incubation at 28 °C [29].

## 2.3. Identification of Rhizobium Host Compliance with Cultivars

Seeds of 10 representative soybean varieties from Heilongjiang Province were sterilized via chlorine fumigation. The seeds were planted under aseptic conditions and inoculated with the four rhizobacterial strains obtained from the screening and the control *Sinorhizobium fredii* (*S. fredii*) HH103, and each plant was inoculated with 2 mL of bacterial sap with a concentration OD$_{600}$ of 0.2 [30]. Each treatment group was replicated 10 times. After 30 days of incubation, the number of nodules, nodule sizes, nitrogenase activity, and dry weights of the nodules were investigated.

## 2.4. Data Collection

For the nodulation experiments, plants were inoculated with the rhizobia by immersing the roots in a bacterial solution at an OD$_{600}$ of 0.2 for ten seconds. At 28 days post-inoculation (dpi), the nodule numbers were counted, and the nodule dry weights were obtained. Nodules appeared on both the main roots and on some lateral roots; all nodules were considered for the analysis. Ten independent biological repeats were performed.

## 2.5. Statistical Analysis

In this study, the nodulation experiments were replicated at least ten times. After 28 days of rhizobial inoculation, the soybean plants were dug out, and the roots were carefully washed with running water until there was no soil on the surfaces. We used tweezers to remove all the nodules on the roots and count them. After counting, the clean,

fresh nodules were baked in an oven at 75 °C for 48 h to evaporate all the water from the nodules, and then the dry weights of the nodules in the different treatment groups were determined [14]. Data collected from the different treatment groups were included in the statistical analyses [31]. In this study, we used a one-way ANOVA to analyze the Suinong 14 inoculated with four different rhizobia and a two-way ANOVA to analyze the data of the different soybean germplasm resources inoculated with four different rhizobia using the SPSS 27.0 statistical software package.

## 3. Results

### 3.1. Isolation and Identification of Rhizobia from Different Regions

The rhizobia isolated from four regions, Heilongjiang, Shaanxi, Yunnan, and Guangdong Provinces, were named HLNEAU1, HLNEAU2, HLNEAU3, and HLNEAU4, respectively, based on their latitudinal geographic characteristics. Monoclonal strains with the rhizobium morphological characteristics were obtained and observed on YMA-medium plates (Figure 1A–D). The purified rhizobium clone phenotype was round or oval, with a moist surface, and opaque, milky white, and slightly raised. This phenotype confirm that the isolated single clones were rhizobia. Based on the results of the PCR amplification, the sequencing of strain 16S rRNA (Figure 1E), and the plotting of the phylogenetic tree with known sequences (Figure 1F), it was shown that the 16S rRNA of the conserved sequence of HLNEAU1 had 99% similarity to the slow-growing strain *Bradyrhizobium japonicum* (*B. japonicum*); the 16S rRNA of HLNEAU2 had 96% similarity to the 16S rRNA of *S. fredii* and *Sinorhizobium* sp. (*S.* sp.); the HLNEAU3 conserved-sequence 16S rRNA had 98% similarity to *S. fredii*; and HLNEAU4 had 97% similarity to *S. fredii* and *S.* sp. The isolated strain HLNEAU1 in this study was a slow-growing rhizobium, and the other three were fast-growing rhizobia.

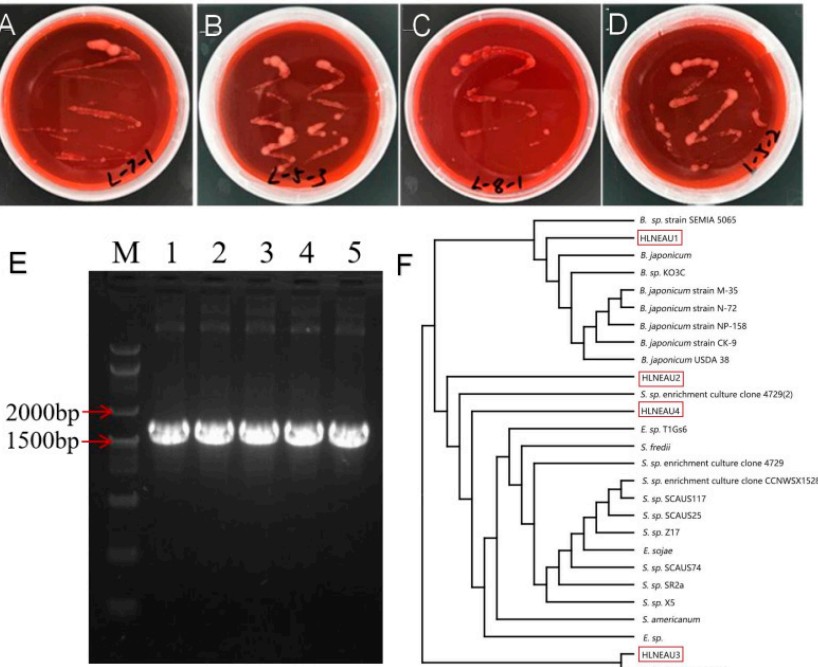

**Figure 1.** The strains HLNEAU1, HLNEAU2, HLNEAU3, and HLNEAU4 isolated from four different regions were identified as rhizobia. (**A–D**) Colony characteristics of HLNEAU1, HLNEAU2, HLNEAU3, and HLNEAU24 in YMA plate. (**E**) PCR identification of Rhizobium 16S rRNA. M represents markers. Numbers 1–5 represent 16S rRNA of HLNEAU1, HLNEAU2, HLNEAU3, HLNEAU4, and HH103, respectively. (**F**) Phylogenetic tree of 16S rRNA sequence of Rhizobium strains from different regions.

### 3.2. Identification of Rhizobium Antibiotic Resistance

3.2.1. Antibiotic Resistance Identification of Rhizobia

According to the antibiotic resistance identification in the rhizobia (Figure 2A–D), the four rhizobia were non-resistant to the common antibiotics gentamicin (Gent), kanamycin (Km), tetracycline (Tet), rifampicin (Rif), and spectinomycin (Spe).

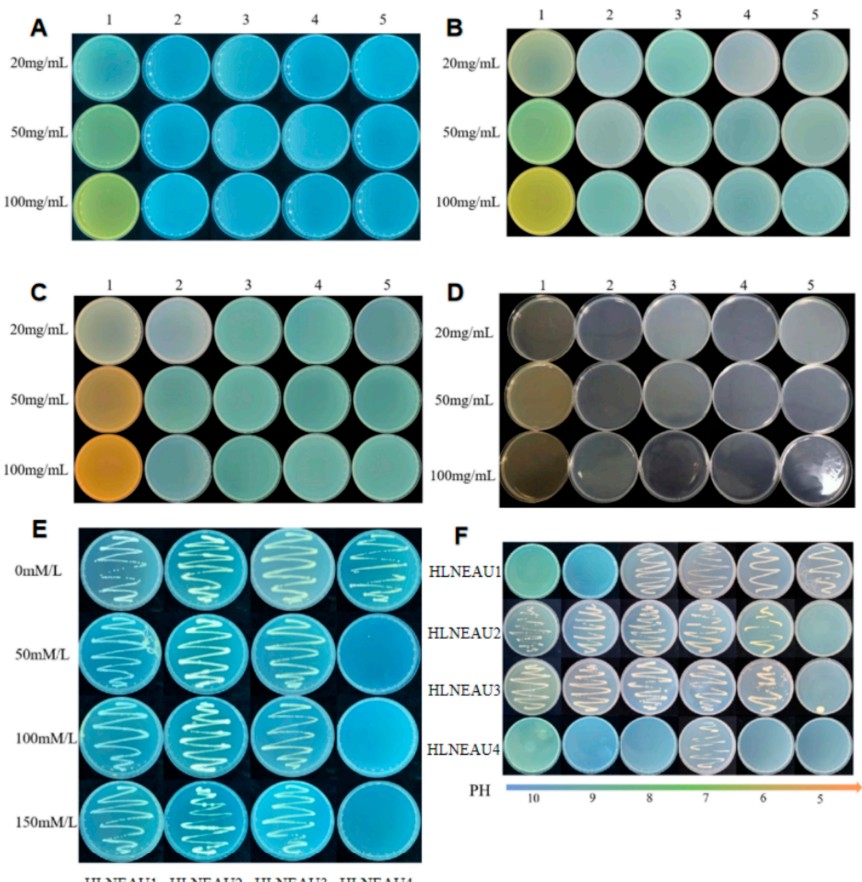

**Figure 2.** Antibiotic screening and salt-Resistance, alkali-resistance and acid-resistance identification of isolated strains. (**A–D**) Screening for resistance at different concentrations of HLNEAU1,HLNEAU2,HLNEAU3,HLNEAU4. Numbers 1–5 represents Rifampicin (Rif), Kanamycin (Km), Spectinomycin (Spe), Gentamicin (Gent) and Tetracycline (Tet), respectively. (**E**) Identification of salt resistance of isolated strains. (**F**) Identification of alkali-resistance and acid- resistance of isolated strains.

3.2.2. Salt Resistance, Alkali Resistance, and Acid Resistance Identification

HLNEAU4 only grew on the medium with the 0 mmol/L NaCl concentration, while HLNEAU1, HLNEAU2, and HLNEAU3 grew normally on the medium with the 150 mmol/L NaCl concentration (Figure 2E). HLNEAU4 could only grow on a neutral medium at a pH of 7, could not grow on any other pH medium, was neither acid- nor alkali-resistant, and only survived normally in neutral soil at a pH of 7. HLNEAU1 could grow on the medium with a pH of 5–8, indicating that this strain was extremely resistant to acid, weakly resistant to alkali, and could grow normally in acidic soil. HLNEAU2 and HLNEAU3 could grow on the medium with a pH of 6–10. These two rhizobia had strong alkali resistances and weak acid resistances and could grow normally in alkaline soil, showing good salinity resistances (Figure 2F).

### 3.3. Identification of Nodulation Phenotypes of Isolated Rhizobia and Compatibilities with Cultivated Soybean Varieties

The four isolated rhizobium strains and the blank control magnesium sulfate were inoculated into Suinong 14. After 28 days of culturing in nitrogen-deficient nutrient solution, the nitrogen-fixing phenotype was studied, including the nodule number, nodule size, and nodule dry weight, in order to identify the nitrogen-fixing abilities of the isolated strains. Because of the symbiotic nitrogen fixation, the plants inoculated with the isolated rhizobia had obviously greener leaves than the control plants. In addition, the nodules produced by all four rhizobia were effective nodules with a pink color in the cross sections. HLNEAU 1 and HLNEAU 4 had significantly more nodules than HLNEAU 2 and HLNEAU 3, while the nodules of HLNEAU 2 and HLNEAU 3 were larger (Figure 3). These results confirm that the isolated strains can inoculate soybean and fix nitrogen efficiently.

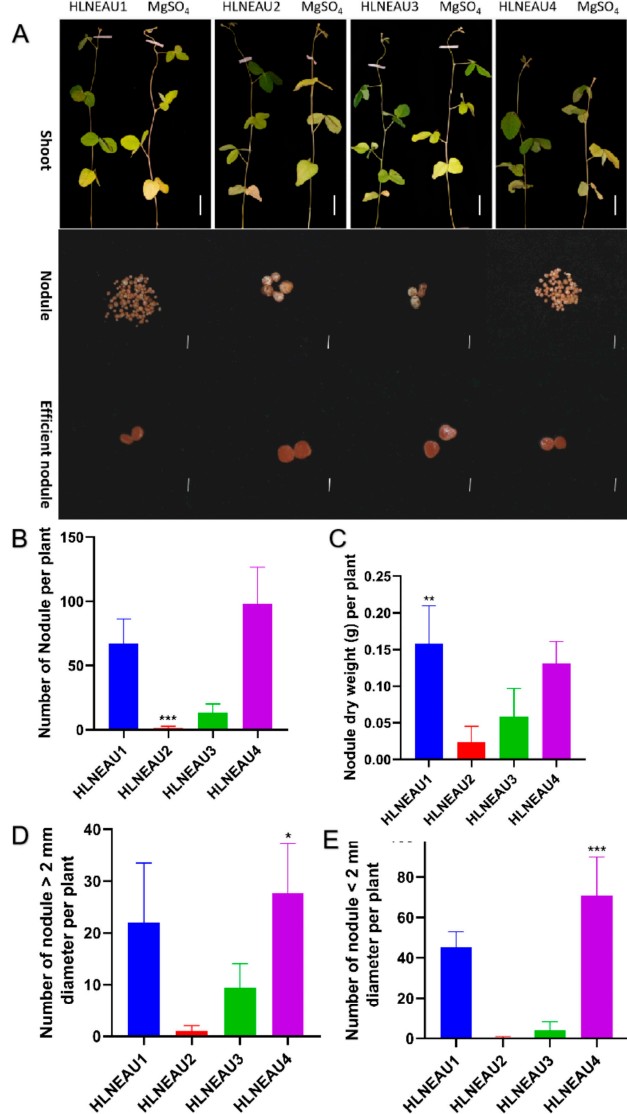

**Figure 3.** The isolated strains can inoculate soybean and fix nitrogen efficiently. (**A**) Aboveground phenotypes of HLNEAU1, HLNEAU2, HLNEAU3, HLNEAU4, and control MgSO4 inoculated with Suinong 14. The color of the nodule cut surfaces produced by HLNEAU1, HLNEAU2, HLNEAU3, and HLNEAU4 inoculated with Suinong 14 was pink; shoot scale bars represent 5 cm; nodule scale bars represent 2 mm. (**B**) Histogram of the number of nodules per plant. (**C**) Histogram of the nodule dry weight per plant. (**D**,**E**) Histograms of the numbers of nodules > 2 mm in diameter and ≤2 mm in diameter per plant, respectively. * Indicates significant difference at $p < 0.05$; ** indicates significant difference at $p < 0.01$; *** indicates significant difference at $p < 0.001$.

### 3.4. Identification of Rhizobium Host Compliances with Cultivars

Ten representative soybean germplasms were selected from six accumulated temperature areas in Heilongjiang Province, and the isolated rhizobia were inoculated to identify the compatibilities of the rhizobia and soybean germplasms. The inoculation of 10 germplasms with the rhizobia showed that HLNEAU 1 and HLNEAU 4 induced more nodules; thus, the plants inoculated with HLNEAU 1 and HLNEAU 4 had heavier nodule dry weights (Figure 4A,B). Regarding the nodule size phenotype, the numbers of nodules induced by HLNEAU1 and HLNEAU4 were greater than those induced by HLNEAU2 and HLNEAU3, both for large nodules (nodule diameters > 2 mm) and small nodules (nodule diameters $\leq$ 2 mm) (Figure 4C,D). Among them, Dongnongdou 252 inoculated with HLNEAU4 produced the highest number of rhizomes that were >2 mm in diameter, while Heinong 56 inoculated with HLNEAU4 produced the highest number of rhizomes that were $\leq$2 mm in diameter. The nitrogenase activities of the nodules produced from the soybean germplasms inoculated with the four rhizosphere bacteria were analyzed (Figure 4E). We found that Hefeng 50 had good compatibility with the four rhizosphere bacteria. Compared with the other three rhizobia strains, Suinong 14 inoculated with HLNEAU2 produced the highest nitrogenase activity, which indicated that Suinong 14 and HLNEAU2 had better compatibility. The inoculation of Heihe 17 with HLNEAU3 resulted in the lowest nitrogenase activity in the root nodules, which proved that the compatibility between Heihe 17 and HLNEAU3 was low. These results show that the isolated rhizobia can successfully establish symbiotic relationships with soybean, but there was also host specificity.

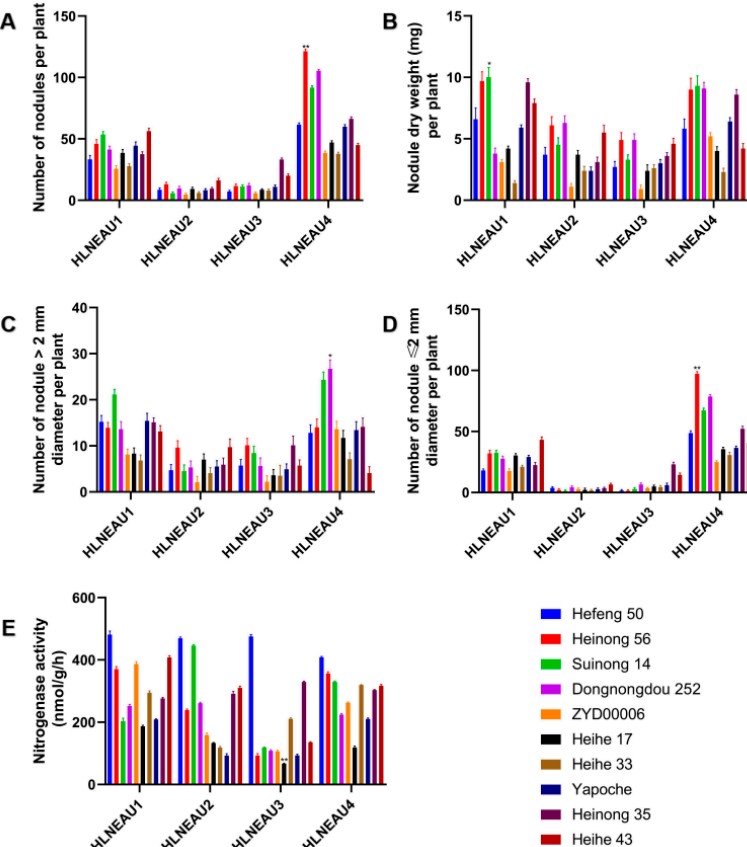

**Figure 4.** Phenotypic identification of 10 representative soybean germplasms inoculated with four isolates of Rhizobium. (**A**) Histogram of the number of nodules per plant. (**B**) Histogram of the nodule dry weight per plant. (**C,D**) Histograms of the numbers of nodules > 2 mm and $\leq$2 mm in diameter per plant, respectively. (**E**) Histogram of nitrogenase activity. * Indicates significant difference at $p < 0.05$; ** indicates significant difference at $p < 0.01$.

## 4. Discussion

### 4.1. Application Analysis of Rhizobia from Different Areas

As the demand for soybeans continues to increase, so does the need for soybean acreage. However, corn and rice occupy most of the land resources, necessitating the development of some saline–alkali or acidic land for planting. The saline–alkali land area in China is about 99.13 million hectares, accounting for about 5% of the available land area [32,33], the utilization of which will greatly improve the supply of agricultural products. For example, about 23% of the land in Inner Mongolia is saline land, covering a comprehensive area of up to 7.63 million hectares [34]. In recent years, many scientists have conducted a significant amount of research on the negative effects of salt stress on soybeans. As a moderately salt-resistant crop, salt-resistant cultivars have also appeared owing to years of research and cultivation. Lu et al. used the saline direct-seeding method to screen and obtain 20 varieties with good overall saline tolerance performance, especially the 'ShengYu24', 'ShaanBeans24', 'ZhongZuo12', 'ShengBeans147', 'HZE19-64', 'LONGBeans655-2', and 'Ji21YG105' varieties [35]. Planting saline-tolerant soybean varieties on saline land and replacing chemical nitrogen fertilizers with saline-tolerant rhizobacteria for nitrogen fixation in soybeans increase soybean yields by expanding the soybean-planting area.

Stress-resistant rhizobia have significant potential in agricultural production. Li et al. isolated the drought-resistant rhizobacterium 'QHCD11' from the root nodules of faba beans growing in an arid agricultural area in the Qinghai–Tibet Plateau, and they found that the inoculation of faba beans with QHCD11 was an environmentally friendly strategy for mitigating drought stress in arid and semi-arid crops [36]. Rhizobium STDF-Egypt19 isolated from the root nodules of faba beans grown in sludge-contaminated fields in Upper Egypt were used as an inexpensive and efficient bioremediation technology for the removal and recovery of heavy-metal ions from aqueous solutions [37]. Native rhizobacteria isolated from the highly saline soil of Vargara are highly salt-tolerant and can enhance the productivity of legumes cultivated in extreme environments [38]. Rhizobial agents are also widely used. *Bradyrhizobium japonicum* shows a slow growth in culture and has been extensively used to produce liquid and solid bioinoculants for application in seeds before sowing [39]. In this study, three strains of rhizobia from different regions that were resistant to 150 mM/L were found via the growth statuses on the medium with different salt concentrations (0 mM/L, 50 mM/L, 100 mM/L, and 150 mM/L), which can be used in the salt-resistant soil of most salt-resistant cultivars. Rhizobium strains were isolated from chickpea nodules, which could grow well at pH values of 6 and 7, temperatures from 28 °C to 30 °C, and a salt concentration of 1% [40]. The *Bradyrhizobium* strain RJS 9-2 was able to grow with a 0.35 M NaCl treatment and promote *Stylosanthes guianensis* growth [41]. The three strains of rhizobia found in this experiment were normal in 150 mM/L, having reached the application level in salt resistance. These isolated strains were derived from the nodules of Suinong 14, and they exhibited varying degrees of acid and alkali resistances. The results of the present study indicate that the four isolated rhizobacterial strains can help legumes to fix nitrogen and increase their yields under extreme conditions.

### 4.2. The Effects of Rhizobia on the Environment

There are many kinds of rhizobia, among which *S. fredii* and *B. japonicum* are two different groups. In early studies, it was found that *S. fredii* is a fast-growing and acid-producing soybean rhizobium [42], while *B. japonicum* is a slow-growing and alkali-producing soybean rhizobium [43]. Because of these physiological characteristics, the pH of the environment has a substantial impact on the survival of rhizobia. For example, the black soil from the cultivated land in northern China, where *B. japonicum* exists, has more humus and a dark color and is loose and mostly acidic soil. However, the central and western regions of China have mostly alkaline soil, with a yellow or red color, and most rhizobacteria in these areas are fast-growing rhizobacteria [44], which is also consistent with the results of the rhizobia isolated in this experiment, in which slow-growing rhizobia were isolated from soil from Heilongjiang Province, and fast-growing rhizobia were isolated from soil

from Xi'an, Shaanxi Province, Guangzhou, Guangdong Province, and Yunnan Province. Furthermore, in the pH resistance determination, the isolated strains in this study could grow on the medium at pH values from 5 to 8, showing acid resistance and slight alkali resistance, while the strains from Yunnan and Shaanxi could grow on the medium at pH values from 6 to 10, showing alkali resistance and slight acid resistance. According to these results, if we want to isolate rhizobia with special resistance characteristics, using special soil from different areas and more arrest hosts is a good strategy.

*4.3. Effects of Compatibility between Soybean and Rhizobacteria on Symbiotic Nitrogen Fixation*

Numerous studies have shown that the symbiotic host compliance of rhizobacteria and soybean is complex, with multiple regulators acting together. Factors such as antibiotics, signaling molecules, synergistic metabolism, and physical interactions can affect the host–rhizobium compatibility. The rhizobial type III secretory protein NopP was identified as the determinant of the symbiotic incompatibility with the Rj2 soybean [45]. The symbiosis of soybean with *Bradyrhizobium diazoefficiens* USDA110 is better than that with *Bradyrhizobium elkanii* USDA94 [46]. In practice, different rhizobia can be selected for compound applications according to different soybean varieties, which may greatly improve the utilization efficiency of rhizobia. This study provides effective information for the use of rhizobia in these excellent cultivars. More detailed experiments are required to elucidate these questions.

**5. Conclusions**

In this study, four rhizobium strains, HLNEAU1, HLNEAU2, HLNEAU3, and HLNEAU4, were isolated and identified from the root nodules of soybeans grown at different latitudes. Acid, alkali, and salt tolerance tests of the four rhizosphere bacterial strains were carried out, and it was found that the four strains showed different degrees of acid, alkali, and salt tolerances. The nodulation and nitrogen fixation phenotypes of the four isolates and compatibility analyses with representative soybean germplasm resources from Northeast China showed that all four rhizobacterial strains have good nodulation and nitrogen fixation abilities and show different degrees of symbiotic relationships with different soybean germplasms. The results of this study provide a theoretical basis for the use of salinity-tolerant rhizobacteria as an alternative to chemical nitrogen fertilizers to improve soybean yields in saline soils.

**Author Contributions:** Conceptualization, D.X., H.N. and Q.C.; methodology, T.Y., H.L., X.W. (Xiaodong Wu) and G.Z.; software, W.T. and X.H.; validation, Z.Z., X.W. (Xiaohan Wang) and Y.G.; formal analysis, Y.S.; investigation, T.Y., J.L. and W.T.; resources, X.W. (Xiaodong Wu); data curation, X.Z. and S.T.; writing—original draft preparation, T.Y., H.L. and H.N.; writing—review and editing, D.X. and H.N.; visualization, Y.S. and T.Y.; supervision, D.X. and H.N.; project administration, T.Y. and Q.C.; funding acquisition, D.X., H.N. and Q.C. All authors have read and agreed to the published version of the manuscript.

**Funding:** This work was funded by the National Natural Science Foundation of China (grant numbers 32070274, 31771882, 32072014, and U20A2027), the Postdoctoral Foundation of China (2023MD734142), and the Postdoctoral Foundation of Heilongjiang Province (LBH-Z22080).

**Institutional Review Board Statement:** Not applicable.

**Informed Consent Statement:** Not applicable.

**Data Availability Statement:** Data available on request from the authors.

**Conflicts of Interest:** The authors declare no conflicts of interest.

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
