# Peer review of "Isolation and Identification of Salinity-Tolerant Rhizobia and Nodulation Phenotype Analysis in Different Soybean Germplasms"

_cimb, doi:10.3390/cimb46040209_

Round 1

Reviewer 1 Report

Comments and Suggestions for Authors

Manuscript entitled ‘’Isolation and Identification of Salinity-tolerant Rhizobia and Phenotypic analysis of Nodulation in Different Soybean Germplasm’’ investigated the four strains of rhizobia (HLNEAU1, HLNEAU2, HLNEAU3, and HLNEAU4) were isolated and identified from the nodules of soybean. The soybeans that used for the current research were planted in soils at different latitudes in China. There are some comments, please see them below:

There are some grammatical and punctual mistakes in the text, so moderate English editing is required.

Also, between references and words should be space, for example in Line 37:

crops[1] -- > should be crops [1]. Please check it for whole the text.

Abstract

The abstract section is not clear. Abstract should be giving a point of view to readers. The method of the abstract should be clear and brief, but in the current research it's unclear and long. I would suggest that the authors dd the rhizobia strains name (HLNEAU1, HLNEAU2, HLNEAU3, and HLNEAU4) in this section.

On the other hand, result is not sufficient and Hypothesis on the research is missing. Please rewrite the abstract section.

Introduction

Line 50-51: ‘’ Related studies have played a role in advancing the development, promotion, and utilization of rhizobium agents’’ -- > bring some of them, compare them with your research and improve the introduction. What is the gap of knowledge in previous research? How the current research can filling this gap?

Introduction should be expanded; The goal and novelty of the research is missing. Please bring relevant references and compare with your research and highlight the novelty of your research.

Material and method

Identification of Rhizobium Host Compliance with Cultivars is well written. But the method of collecting data after harvest isn’t clear (Section 2.5. Data collection). Please expand it and add relevant reference(s) for this section

2.6. Statistical Analysis -- > one way ANOVA or two-way ANOVA?

Why the authors use LSD for comparison between treatments? LSD used for comparing treatments with a control. Which treatment is as a control?

Results

All the figures should refer in the text. Figure 4 contains (A-E), the authors refer them in the text in line 211 -- > ‘’ ...  17 and HJNEAU 3 was low (Figure 4).’’ Please correct it.

Discussion

This section is not sufficient. The authors should compare the result of the current research with previous ones. All the discussion is three paragraphs. The discussion For a research with four strains of rhizobia with identification on resistance, acid and saline tolerance and nitrogen fixation capacity, is not enough.

Conclusion

The conclusion is not impressive. It's repetition of the results! What is the author suggestion for the further research? How your results filled the gap of knowledge?

I would suggest that the authors rewrite this section.

References

References should be added based on the expansion of discussion section!

Comments on the Quality of English Language

There are some grammatical and punctual mistakes in the text, so moderate English editing is required.

Author Response

Dear Editors and Reviewers:

Thank you for the referee’s comments concerning our manuscript entitled‘Isolation and Identification of Salinity-tolerant Rhizobia and Phenotypic analysis of Nodulation in Different Soybean Germplasm’. We have studied their comments carefully and have made correction which we hope meet with their approval. We hope that the revision is acceptable and look forward to hearing from you soon. Thank you and best regards.

Yours sincerely,

Dawei Xin and Hejia Ni

Answer to reviewer1:

  1. There are some grammatical and punctual mistakes in the text, so moderate English editing is required.

Answer: Thank you for your suggestion, we revised the manuscript as your suggestion.

  1. Also, between references and words should be space, for example in Line 37:crops[1] -- > should be crops [1]. Please check it for whole the text..

Answer: Thank you for your suggestion. We have revised the manuscript as your suggestion, we checked the full text and left a space between the references and the words.

  1. The abstract section is not clear. Abstract should be giving a point of view to readers. The method of the abstract should be clear and brief, but in the current research it's unclear and long. I would suggest that the authors add the rhizobia strains name (HLNEAU1, HLNEAU2, HLNEAU3, and HLNEAU4) in this section.On the other hand, result is not sufficient and Hypothesis on the research is missing. Please rewrite the abstract section.

Answer: Thank you for your suggestion, we rewrite the abstract section and added the rhizobia strains name in this section.

  1. Line 50-51: “Related studies have played a role in advancing the development, promotion, and utilization of rhizobium agents”-- > bring some of them, compare them with your research and improve the introduction. What is the gap of knowledge in previous research? How the current research can filling this gap?

Answer: Thank you for your suggestion, we added the statements related to utilization of rhizobium agents in the introduction section as your suggestion, and adds the implications of what this study has on complementing the gaps in previous research.

  1. Introduction should be expanded; The goal and novelty of the research is missing. Please bring relevant references and compare with your research and highlight the novelty of your research.

Answer: Thank you for your suggestion. We revised and supplemented the introduction according to your suggestion. 

  1. Identification of Rhizobium Host Compliance with Cultivars is well written. But the method of collecting data after harvest isn’t clear (Section 2.5. Data collection). Please expand it and add relevant reference(s) for this section.

Answer: Thank you for your suggestion, we expanded the method of collecting data after harvest and added relevant reference for this section as your suggestion.

  1. 6.Statistical Analysis -- > one way ANOVA or two-way ANOVA?

 Answer: Thank you for your suggestion. We added a detailed description of the statistical analyses. In this study, one-way ANOVA was used to analyze Suinong 14 inoculated with four different rhizobia, and two-way ANOVA was used to analyze the data of different soybean germplasm resources inoculated with four different rhizobia..

  1. Why the authors use LSD for comparison between treatments? LSD used for comparing treatments with a control. Which treatment is as a control?

Answer: Thank you for your suggestion. Multiple comparisons of data as a method of analysis were not carried out in this work, and we have removed irrelevant descriptions in the section 2.5 Statistical analyses.

  1. All the figures should refer in the text. Figure 4 contains (A-E), the authors refer them in the text in line 211 -- > ‘’ ...  17 and HJNEAU 3 was low (Figure 4).’’ Please correct it.

Answer: Thank you for your suggestion. We have referenced A-E of Figure 4 in the appropriate places as your suggestion. 

  1. The authors should compare the result of the current research with previous ones. All the discussion is three paragraphs. The discussion For a research with four strains of rhizobia with identification on resistance, acid and saline tolerance and nitrogen fixation capacity, is not enough.

Answer: Thank you for your suggestion, we revised the manuscript as your suggestion.

  1. The conclusion is not impressive. It's repetition of the results! What is the author suggestion for the further research? How your results filled the gap of knowledge? I would suggest that the authors rewrite this section.

Answer: Thank you for your suggestion, we have rewritten the conclusion section based on your suggestions.

  1. References should be added based on the expansion of discussion section!

Answer: Thank you for your suggestion, we revised the references as your suggestion.

Reviewer 2 Report

Comments and Suggestions for Authors

In the present manuscript, titled "Isolation and Identification of Salinity-tolerant Rhizobia and 2 Phenotypic analysis of Nodulation in Different Soybean 3 Germplasm." the authors conducted an interesting study that provides new insights into the theoretical basis of nitrogen fixation in soybean in saline soils using acid-, salt- and alkali-tolerant rhizobia. After careful selection of rhizobia based on their environmental adaptability, the authors came to the convincing conclusion that they can be inoculated for their genetic compatibility with local soybean and legume varieties.

The experiment was well performed and the molecular biology methods adopted, such as PCR amplification and high-throughput sequencing for microbial community determination, are reliable and effective.

The results obtained and the conclusions reached by the authors leave little room for comments

The work is original and can be accepted if the authors respond to the following reviewer's observation

The authors cite (also specifying geographic regions) vast arable areas of China damaged ecologically and environmentally, without indicating the causes of either the accumulation of acid or base salts or their chemical nature. I suggest they clarify more in the introduction and accompany the paper with chemical analysis of the affected soils.

As is well known, root colonization occurs through the production of auxin by rhizobia and the subsequent activation of expansins, which loosen the cellulose meshwork of the wall promoting entry of the bacterium into the host cell. I ask the authors to further clarify in the introduction whether auxin production by rhizobia can be substrate-dependent.

To be more clear, having shown that certain strains of rhizobia grow better in acidic, basic or saline soils does not necessarily mean that they are able to fix nitrogen well under those certain conditions.

In particular, I ask the authors whether indolacetic acid can maintain the same efficacy in an alkaline soil and whether it is not the slightly acidic soils that chemically favor this process, for three orders of reasons:

1) Indolacetic acid would lose its acidic characteristic in an alkaline environment.

2) The fixation of a nitrogen molecule presupposes the consumption of 6 protons, which can hardly be sequestered by an alkaline environment.

3) Ammonia produced by nitrogen fixation, being an alkaline pH gas, could partly leak into the atmosphere with considerable loss of a share of inorganic nitrogen from the soil.

Having said that, it cannot be ruled out that rhizobium fixes nitrogen well in alkaline symbiosis with leguminous plants; it would be enough to do some tests under controlled conditions in a greenhouse and determine the root colonization index.

I invite the authors more clarification on this aspect.

Line 233, Write 150 mM or 150 mmol /L

Line 235 perform the same correction.

Comments on the Quality of English Language

English language is acceptable, deserves only moderate revision.

Author Response

Dear Editors and Reviewers:

Thank you for the referee’s comments concerning our manuscript entitled‘Isolation and Identification of Salinity-tolerant Rhizobia and Phenotypic analysis of Nodulation in Different Soybean Germplasm’. We have studied their comments carefully and have made correction which we hope meet with their approval. We hope that the revision is acceptable and look forward to hearing from you soon. Thank you and best regards.

Yours sincerely,

Dawei Xin and Hejia Ni

Answer to reviewer2:

  1. The authors cite (also specifying geographic regions) vast arable areas of China damaged ecologically and environmentally, without indicating the causes of either the accumulation of acid or base salts or their chemical nature. I suggest they clarify more in the introduction and accompany the paper with chemical analysis of the affected soils.

Answer: This is good suggestion. We added relative description in the introduction section.

  1. As is well known, root colonization occurs through the production of auxin by rhizobia and the subsequent activation of expansins, which loosen the cellulose meshwork of the wall promoting entry of the bacterium into the host cell. I ask the authors to further clarify in the introduction whether auxin production by rhizobia can be substrate-dependent.

Answer: This is good suggestion. We added the relative introduction of auxin underlying symbiosis.

  1. To be more clear, having shown that certain strains of rhizobia grow better in acidic, basic or saline soils does not necessarily mean that they are able to fix nitrogen well under those certain conditions.

Answer: This is a good comment. In this study we use the arrest host isolated the rhizobium had have a compatible capacity with host soybean. And we also analysis the nitrogen fixation enzyme activity. The acidic and basic resistance are the additional character we wish can by use in the future.

  1. In particular, I ask the authors whether indolacetic acid can maintain the same efficacy in an alkaline soil and whether it is not the slightly acidic soils that chemically favor this process, for three orders of reasons:

1) Indolacetic acid would lose its acidic characteristic in an alkaline environment.

2) The fixation of a nitrogen molecule presupposes the consumption of 6 protons, which can hardly be sequestered by an alkaline environment.

3) Ammonia produced by nitrogen fixation, being an alkaline pH gas, could partly leak into the atmosphere with considerable loss of a share of inorganic nitrogen from the soil.

Answer: This is an interesting question. We wish we can answer this question in future by specificity experiment. In this study, our work is focus on the rhizobium isolation and the resistance character analysis.

  1. Having said that, it cannot be ruled out that rhizobium fixes nitrogen well in alkaline symbiosis with leguminous plants; it would be enough to do some tests under controlled conditions in a greenhouse and determine the root colonization index.

Answer: This is a very valuable suggestion for us work. We will design relative experiment to detect this scientific question in future. Thank you very much, it is a very kindly recommendation.

Round 2

Reviewer 1 Report

Comments and Suggestions for Authors

I reviewed the authors' comprehensive answers and efforts to revise the text. The paper has improved significantly after implementing all the suggestions.